# Uptake of diabetic retinopathy screening at a secondary level facility in Malawi

**Thokozani Zungu**[1,2]✉*, **Shaffi Mdala**[1,2]✉, **Petros Kayange**[1,2], **Elizabeth Fernando**[2], **Halima Twabi**[3], **Arnold Jumbe**[4], **Johnstone Kumwenda**[1,2], **Adamson Muula**[1]

**1** Kamuzu University of Health Sciences, Blantyre, Malawi, **2** Queen Elizabeth Central Hospital, Blantyre, Malawi, **3** Department of Mathematical Sciences, University of Malawi, Zomba, Malawi, **4** Thyolo District Hospital, Thyolo, Malawi

✉ These authors contributed equally to this work.
* tzungu@kuhes.ac.mw

**Data Availability Statement:** The data for this this project is fully available without restriction through the following link DOI: https://doi.org/10.6084/m9. figshare.23710023.v1.

## Abstract

Diabetic retinopathy (DR) is a common microvascular complication of long-standing diabetes mellitus (DM). DR screening is a cost-effective intervention for preventing blindness from DR. We conducted a cross-sectional study to investigate the uptake and the predictors of uptake of annual DR screening in an opportunistic DR screening programme at a secondary-level diabetes clinic in Southern Malawi. Consecutive patients were interviewed using a structured questionnaire to record their demographic characteristics, medical details and data regarding; the frequency of clinic visits, knowledge of existence of DR screening services and a history of referral for DR screening in the prior one year. Univariate binary logistic regression was used to investigate predictors of DR screening uptake over the prior one year. Explanatory variables that had a P-value of < 0.1 were included into a multivariate logistic regression model. All variables that had a p-value of <0.05 were considered to be statistically significant. We recruited 230 participants over three months with a median age of 52.5 years (IQR 18–84) and a median duration of diabetes of 4 years (IQR 1–7). The average interval of clinic visits was 1.2 months (SD ± 0.43) and only 59.1% (n = 139) of the participants were aware of the existence of diabetic retinopathy screening services at the facility. The uptake for DR screening over one year was 20% (n = 46). The strongest predictors of uptake on univariate analysis were awareness of the existence of DR screening services (OR 10.05, P <0.001) and a history of being referred for DR screening (OR 9.02, P <0.001) and these remained significant on multivariable analysis. Interventions to improve uptake for DR screening should promote referral of patients for DR screening and strengthen knowledge about the need and availability of DR screening services.

## Introduction

**More than 24 million adults aged 20–79 in Africa have diabetes mellitus (DM) and this figure is estimated to increase to 33 million by 2030 and 55 million by 2045 [1].** Diabetic retinopathy (DR) is a common complication of DM among adults and its prevalence in sub-

**Funding:** This study was funded by the Non-communicable Diseases Building Research Capacity, Implementation and Translation Expertise consortium (NCD BRITE) mentored research grant to TZ for the period 2020 to 2022. The NCD BRITE consortium is supported by the National Heart, Lung, and Blood Institute of the National Institutes of Health under grant number 5U24HL136791 to AM. The funding covered participant compensation fees, communication costs and transportation costs for the study team. The funders had no role in study design, data collection and analysis, decision to publish, or preparation of the manuscript.

**Competing interests:** The authors have declared that no competing interests exist.

Saharan Africa is increasing. It is estimated that a third of persons living with DM in the region have DR and 10% have sight-threatening DR [2]. DR is usually asymptomatic in its early stages and it may thus not be noticed until a patient's vision is affected and the risk of blindness is high [3, 4].

There is strong evidence that DR screening is a cost-effective intervention for the prevention of visual impairment and blindness from diabetes [5]. However, DR screening uptake is often low even in well-resourced settings [6–8]. Some of the factors that have been reported to be associated with non-attendance for DR screening include younger age, lack of ocular symptoms, fear and low socioeconomic status. Other factors include long duration of diabetes, type of diabetes (with people with type 1 DM being less likely to attend screening), lack of awareness of DR and its risks and poor access to DR screening centres [9, 10]. There is also evidence that a referral by a primary health care professional for a patient to attend DR screening encourages screening attendance [10–13].

For high income countries, systematic DR screening is utilised to achieve a high coverage of DR screening [14]. This is organised at the population level and it utilises a register of people living with diabetes to manage a call and recall system to ensure that all eligible people are offered or reminded of an appointment for screening [13]. However, most countries in Sub Saharan Africa lack the resources required to set up and maintain systematic DR screening programmes [15]. Thus, opportunistic screening for DR is often utilised in most screening programmes in the region.

Opportunistic screening involves providing DR screening to patients with diabetes upon their presentation to a health facility to seek DM care [15]. Considering the shortage of health facilities, personnel skilled in DR screening and the poor accessibility to facilities offering this service, diabetic retinopathy may go unnoticed in most patients. Thus most patients present to eye health facilities only after experiencing considerable vision loss [2]. It is thus important to establish interventions that may promote the uptake of DR screening in an opportunistic DR screening programme.

Malawi is a low-income country in Southern Africa with 28 districts and a population of 19.9 million of whom 84% live in rural areas [16]. The prevalence of DM among adults is 2% in rural areas and 3% in urban areas and the Ministry of Health has a Non-communicable diseases policy which includes DM management [17]. In the country's national DR screening programme, ophthalmic clinical officers (OCOs) and optometrists perform opportunistic DR screening using direct ophthalmoscopy with pharmacological pupillary dilatation. This is done among patients attending diabetes clinics in the country's twenty-four district hospitals, which are all secondary-level facilities that serve largely rural populations. Patients identified with treatable DR are referred for treatment to the country's four tertiary level hospitals, which are located in the country's four cities. There are no resources for DR screening in primary level health facilities.

The uptake of DR screening at any level of the health system in Malawi had not been reported prior to this study. The aim of this study was to describe the uptake and factors associated with uptake of DR screening in an opportunistic DR screening programme at a secondary level diabetes clinic in Southern Malawi.

## Materials and methods

### Ethics statement

The study protocol was reviewed and approved by the College of Medicine Research and Ethics Committee, Malawi, P.10/19/2803 and adhered to the tenets of the Declaration of Helsinki for research involving human participants. A participant information sheet was used to explain

about the study's purpose and its procedures during the health education talks that the patients attend in a group before being seen in the clinic. Thereafter, the patients were individually invited to a private room where they were given a chance to ask questions about the study, written informed consent was obtained and data collection interviews were held. The collected data was de-identified upon transcription onto a data collection sheet. All hard copy records were kept in a lockable cabinet and the data was entered into a Microsoft Excel spreadsheet (S1 Table).

All explanations about the study and the informed consent process were done by a study nurse from Queen Elizabeth Central Hospital, which is the referral facility for Thyolo District Hospital.

## Study setting

This was a cross-sectional study that was carried out over 3 months in the diabetes clinic at Thyolo district hospital between the months of October 2020 to December 2020. The diabetes clinic is run on one day per week and about thirty patients with diabetes are seen in each clinic. The hospital is a secondary-level government-run health facility in Southern Malawi and it serves a rural population of about 721,456 people [18]. All healthcare services in public hospitals in Malawi, including DR screening and treatment, are government-funded and are provided at no cost to patients at the point of delivery. Besides ophthalmoscopy, DR screening includes visual acuity checks and anterior segment examination with a slit lamp to screen for other visually significant eye diseases that may affect patients with DM such as glaucoma, cataract and refractive error. In contrast to tertiary hospitals, there are no facilities for checking intraocular pressure or for performing ancillary tests such as perimetry in all the 24 district hospitals. At the time of the study, the diabetes clinic at Thyolo district hospital had a total of 700 patients on its register of patients.

As part of routine care, a group-based health education session is delivered by a nurse during each clinic and it includes information for patients on the need for annual DR screening. The nurse-led diabetes health education is meant to reduce the amount of time that the clinician running the diabetes clinic spends on counselling each patient about the complications of DM and the need for DR screening. DR screening is performed in the eye clinic at the facility and patients attend screening according to the schedule advised to them during the nurse-led group health education sessions.

The patient waiting time is prolonged for not less than one hour for patients undergoing DR screening. This is because they undergo pupillary dilatation with 1% tropicamide eye drops before fundoscopy and they also wait on a queue along with patients awaiting to be attended to for other ocular conditions in the eye clinic.

## Subjects

The study included all consecutive patients aged at least 18 years old who attended the diabetes clinic at the hospital over the study period. The majority of the patients had a relative who had escorted them to the hospital and these signed as witnesses to the informed consent. Patients who had no witnesses were given an appointment to take part in the study on the following week after returning with a witness. Participants or witnesses who were illiterate signed the consent forms with a thumb print and the only criterion for exclusion was a refusal to participate in the study.

## Data collection

A nurse interviewed each participant and ascertained the following data from the history and clinical records of each participant included in the study: age, sex, level of education, history of

hypertension, type of diabetes, duration of diabetes, HIV status and the intervals at which the patient visited the clinic over the past 12 months. The nurse also asked each participant if they were aware of the existence of DR screening services at the facility, whether they had been individually referred by the DM clinician for DR screening within the prior 12-month period and whether they attended DR screening over the prior 12 months. After the interview, the visual acuity of each study participant was measured by an OCO using a Snellen chart.

### Statistical analysis

The data collected was entered and managed using a Microsoft excel spread sheet and statistical analysis was performed using STATA 17. The main outcome variable was the uptake of DR screening (which was defined as the proportion of patients who had a diabetic eye examination over the past one year).

Continuous variables were summarised as means or medians at 95% confidence intervals and univariate logistic regression was used to investigate the association between uptake of DR screening and the following explanatory variables; age, sex, highest level of education attended, type of diabetes, duration of diabetes, awareness about DR screening services, history of referral by a clinician for DR screening, intervals of clinic visits, history of hypertension, HIV status and the presenting visual acuity. Variables that had a p-value of <0.05 were considered to be statistically significant. Explanatory variables that had a p-value of < 0.1 on univariate analysis were included into a multivariate logistic regression model.

## Results

### Participant characteristics

We recruited 230 participants over a three-month period and all eligible participants consented to participate in the study. The median age of the patients' population was 52.5 years (IQR 18–84). The majority of the participants were female (56.5%) and most had attended varying levels of formal education except for 31 participants (13.5%). Most patients had type 2 DM (n = 214, 93%) and 139 patients (60.4%) had a history of hypertension. The median duration of diabetes for the study population was 4 years (IQR 1–7) and the average interval of clinic visits was 1.2 months (SD ± 0.43). Although 59.1% (n = 139) of the population were aware of the existence of diabetic retinopathy screening services at the facility, only 33.9% (n = 78) had been referred by a clinician for DR screening in the prior 12 month period and the uptake of DR screening was 20% (n = 46), Table 1.

### Predictors of uptake of diabetic retinopathy screening

Table 1 also shows the factors associated with DR screening over 12 months. The strongest predictors of uptake of DR screening over one year were being aware of the existence of DR screening services (OR 10.05, 95% CI: 3.46–29.18) and a history of being individually referred by a clinician for DR screening within the year (OR 9.02, 95% CI: 4.30–18.90). Participants who had a diagnosis of hypertension had twice the odds (OR 2.13, 95% CI: 1.04–4.38) of having undergone DR screening in the prior 12 months compared to those who did not have hypertension. There was also an association between DR screening uptake and the duration of diabetes with the odds increasing by 1.07 times each year (P = 0.009). The association between being screened for DR in the past year and increasing age was non-significant (OR 1.03, 95% CI: 1.00–1.05) and there was no significant association with the participants' level of education, type of diabetes and the presenting visual acuity.

**Table 1. Patients' characteristics and association with uptake of DR screening (n = 230).**

| Variable | Summary of characteristics | DR screening in past 12 months (uptake), n (%) | | | |
|---|---|---|---|---|---|
| | n (%) | Screened, n = 46 (20.0) | Not screened, n = 184 (80.0) | OR (95% CI)* | P–value* |
| **Age (mean ±SD)** | 52.5 ± 12.9 | 55.8±12.5 | 51.7±12.9 | 1.03 (1.00–1.05) | 0.057 |
| **Sex** | | | | | |
| Female | 130 (56.5) | 23 (10) | 107 (46.5) | 1 | |
| Male | 100 (43.5) | 23 (10) | 77 (33.5) | 1.39 (0.73–2.65) | 0.320 |
| **Level of education attended** | | | | | |
| No formal education | 31 (13.5) | 3 (1.3) | 28 (12.2) | 1 | |
| Primary school | 122 (53) | 27 (11.7) | 95 (41.3) | 2.65 (0.75–9.40) | 0.131 |
| Secondary school | 60 (26.1) | 12 (5.2) | 48 (20.9) | 2.33 (0.61–8.99) | 0.218 |
| Post-secondary education | 17 (7.4) | 4 (1.7) | 13 (5.7) | 2.87 (0.56–14.73) | 0.560 |
| **Type of diabetes** | | | | | |
| Type 1 | 16 (7) | 4 (1.7) | 12 (5.2) | 1 | |
| Type 2 | 214 (93) | 42 (18.3) | 172 (74.8) | 0.73 (0.22–2.39) | 0.605 |
| **Duration of diabetes (median years, IQR)** | 4 (1–7) | 6 (2–11) | 3 (1–7) | 1.07 (1.02–1.12) | 0.009 |
| **Diagnosis of hypertension** | | | | | |
| No hypertension | 91 (39.6) | 12 (5.2) | 79 (34.4) | 1 | |
| Hypertension | 139 (60.4) | 34 (14.8) | 105 (45.6) | 2.13 (1.04–4.38) | 0.039 |
| **HIV status** | | | | | |
| Non-reactive | 174 (75.7) | 38 (16.5) | 136 (59.2) | 1 | |
| Reactive | 41 (17.8) | 7 (3) | 34 (14.8) | 0.74 (0.30–1.79) | 0.501 |
| Unknown | 15 (6.5) | 1 (0.4) | 14 (6.1) | | |
| **Monthly intervals of clinic visits (mean ± SD)** | 1.2 ± 0.43 | 1.3±0.47 | 1.2±0.42 | 1.79 (0.90–3.56) | 0.098 |
| **Awareness about existing DR screening services** | | | | | |
| Not aware | 94 (40.9) | 4 (1.7) | 90 (39.1) | 1 | |
| Aware | 136 (59.1) | 42 (18.3) | 94 (40.9) | 10.05 (3.46–29.18) | <0.001 |
| **Referral for DR screening in past 12 months** | | | | | |
| Not referred | 152 (66.1) | 12 (5.2) | 140 (60.9) | 1 | |
| Referred | 78 (33.9) | 34 (14.8) | 44 (19.1) | 9.02 (4.30–18.90) | <000.1 |
| **Presenting visual acuity (VA) in the better eye** | | | | | |
| 6/6–6/18 (Normal Vision) | 181 (78.7) | 34 (14.8) | 147 (63.9) | 1 | |
| <6/18–6/60 (Visual impairment) | 28 (12.2) | 5 (2.2) | 23 (10) | 0.94 (0.33–2.65) | 0.907 |
| <6/60–3/60 (Severe Visual impairment) | 13 (5.7) | 4 (1.7) | 9 (3.9) | 1.92 (0.56–6.61) | 0.300 |
| <3/60 –NPL (Blindness) | 8 (3.4) | 3 (1.3) | 5 (2.2) | 2.59 (0.59–11.39) | 0.207 |

*Univariate logistic regression, CI: confidence interval

On multivariate analysis, the only factors that remained as significant independent determinants of uptake of DR screening were awareness of the existence of DR screening services at the health facility (P = 0.021) and being individually referred for DR screening within the year (P < 0.001), Table 2.

## Discussion

There is limited data from sub-Saharan Africa on the uptake of DR screening. A systematic review on the barriers and enablers for access to DR screening services in different income settings identified only three primary studies from low-income countries of which two were from sub-Saharan Africa [19]. The uptake of DR screening was low across the studies and this is in line with the findings from our study where only a fifth of the patients had undergone DR

**Table 2. Factors associated with uptake of diabetic retinopathy screening.**

| Variable | OR (95% C.I) | P–value |
| --- | --- | --- |
| **Age** | 1.02 (0.99–1.05) | 0.263 |
| **Duration of Diabetes** | 1.04 (0.98–1.10) | 0.223 |
| **Awareness about DR screening services** | | |
| Not aware | 1.00 | |
| Aware | 3.98 (1.23–12.90) | 0.021 |
| **Referral for DR screening** | | |
| Not individually referred | 1.00 | |
| Individually referred | 4.80 (2.11–10.96) | <0.001 |
| **Intervals of clinic visits** | 1.40 (0.63–3.10) | 0.409 |
| **Diagnosis of hypertension** | | |
| No Hypertension | 1.00 | |
| Hypertension | 1.04 (0.42–2.56) | 0.934 |

screening in the past year. Our findings are comparable to reports from Tanzania, where the uptake for DR screening at a tertiary eye hospital in the past 12 months was found to be 29% [20]. Similarly, the uptake for DR screening among patients attending 9 diabetes clinics across three counties in Kenya was low, with only 13% having undergone screening in the prior one year [13].

One of the possible reasons for the low uptake of DR screening in our study is the rural setting of the study. Although nurse-led patient education on the need for DR screening is delivered to the diabetes clinic attendants, this does not address the inherent barriers that are experienced by patients living in rural communities, such as limited transportation and access to health care services [21]. Rural residence has not been reported to be a predictor of DR screening uptake in the previous African studies [13, 20]. This could have been due to the low numbers of rural-based patients who had undergone DR screening in the studies. Otherwise it has been reported from China and the United Kingdom that rural populations have lower access to DR screening facilities compared to urban populations and consequently have a lower uptake of DR screening [16, 22].

The longer waiting time that patients experience when they attend DR screening in the general eye clinic after the diabetes clinic is another possible reason for the low uptake of DR screening in our study. Long patient waiting time has been reported to be a deterrent for patients to utilise DR screening services in Africa [23]. A possible intervention to reduce the patient waiting time is the integration of DR screening into the diabetes clinic. This eliminates the need for patients to make a second visit to the eye clinic and ensures that DR screening is offered as part of the continuum of care in the diabetes clinic [24]. We thus suggest studies investigating the feasibility of integrating DR screening into diabetes clinics in secondary-level health facilities.

Other important patient-related factors that have been described to be barriers to accessing DR screening in low income countries include; a lack of knowledge on ocular complications of DM, lack of awareness about the importance of eye examination and lack of knowledge about availability of eye clinics [19]. In our study, being aware of the existence of DR screening services at the hospital was strongly associated with uptake for DR screening. Our findings suggest that health education on the availability of DR screening services has an important role in promoting uptake of DR screening in an opportunistic DR screening model.

Although awareness of the existence of DR screening services was an important predictor of uptake for DR screening in our study, only 59.1% of the participants had knowledge about

the existence of the screening services at the facility. This is despite the observation that the patient population visited the diabetes clinic at average intervals of 1.2 months and that they routinely received group health education on the need for DR screening and the existence of the service. This suggests that the health education model being implemented through group counselling may have a low coverage among the patient population. This warrants research into the feasibility of integrating alternative health education models such as personalised counselling for DR into DM clinics. Personalised health education for DM and DR offered by primary health care workers in a rural setting was demonstrated to be effective in improving uptake for DR screening in India [25]. Another intervention that has been shown to be acceptable in an African setting is health education led by peers who have had DR screening [13, 20]. Thus, we suggest research into the effectiveness of peer-led health education to increase uptake of DR screening.

Besides awareness of the existence of DR screening services at the facility, the only other factor associated with uptake of DR screening on multivariate analysis was being individually referred by a clinician for DR screening and this was the strongest predictor of uptake. This is comparable to findings from Kenya where patients who had been referred for an eye examination had higher odds of having had a fundoscopy in the prior 12 months compared to those who had not been referred [13].

Our findings highlight that it may not be sufficient to provide group education to patients on when and where they should go for DR screening, rather it is also important to ensure that each patient with DM is individually referred by a DM clinician for screening. However, previous research from Africa has shown that less than a fifth of physicians caring for people living with DM refer their patients for DR screening [26]. In line with this, only a third of the participants in our study reported that they had been individually referred for DR screening by a clinician. This highlights a poor linkage between DM care and DR screening services. An intervention that has been suggested to promote both referral and screening for DR is integration of DR services into DM services to ensure that patients access them as one service [27].

Patients that have a longer history of DM (particularly of at least 10 years) have a higher risk of developing sight-threatening DR [28]. Thus it may be expected that they are more likely to be referred for DR screening or to have a higher perceived need for undergoing DR screening. Long duration of DM was reported to be a strong predictor of uptake of DR screening in Kenya, India and among non-indigenous populations in Australia [13, 29, 30]. However, comparable to findings from Tanzania, long duration of DM was not an independent predictor of DR screening uptake in our study [20]. One possible explanation for this is the shorter median duration of DM of 4 years among our patient population.

Comorbidity has been shown to increase health care utilisation in general among patients with DM [31]. In Kenya, comorbid hypertension was a predictor for having ever had an eye examination but not for eye examination within 12 months [13]. From our study, although a diagnosis of hypertension was an independent predictor of uptake of DR screening, this was not significant on multivariate analysis. In addition, HIV infection, another chronic comorbidity, had no association with uptake of DR screening.

Our study has some limitations. Due to the cross-sectional study design, a temporal relationship between the predictors of DR screening uptake and the uptake of DR screening could not be established. In addition, the association between the variables are subject to being significantly influenced by the small number of participants that underwent DR screening. Thus, the observations that age, level of education and visual acuity were not significantly associated with uptake of DR screening should be interpreted with caution. Previous reports from Kenya, Nigeria and mostly high income countries have shown that increasing age and low visual acuity are associated with increased attendance for DR screening [13, 19, 32, 33]. In contrast, low

educational attainment and poor literacy have been reported to be associated with poor knowledge of DR and a lower uptake of DR screening irrespective of a country's income level [19].

Another limitation of our study is that it did not have a qualitative component looking into the patient-related reasons for attendance or non-attendance to DR screening especially among the patients who were aware of the existence of DR services. In addition, due to lack of facilities for performing HbA1C at the study location, we could not investigate the relationship between uptake of DR screening and patients' glycaemic control.

One strength of this study is that it is the first study to report on the uptake of DR screening at secondary level health facility in Malawi and the findings add to the limited body of literature on the uptake of DR screening in low income countries. Our findings are of value to policy makers and programme managers planning to improve diabetic retinopathy screening programmes in similar settings.

## Conclusion

Our study showed that there was a low uptake of annual eye examinations through opportunistic DR screening at a secondary level diabetes clinic. Although awareness of DR screening services and a history of referral for screening were strong predictors of screening uptake, there was low awareness of the existence of DR screening services at the facility and most patients had not been referred for screening. Future research should investigate the feasibility of integrating DR counselling and screening services into routine DM care to promote DR screening uptake by increasing patient awareness and facilitating referrals.

## Supporting information

**S1 Checklist. STROBE statement.**
(DOCX)

**S1 Table. Variables collected as part of the study.**
(XLSX)

## Acknowledgments

We are grateful to Ms Asante Makuta for coordinating the implementation of the project. We are also thankful to Mr David Mtumodzi and Ms Myrnah Pendame for assisting in data collection.

## Author Contributions

**Conceptualization:** Thokozani Zungu, Shaffi Mdala, Petros Kayange, Elizabeth Fernando, Halima Twabi, Arnold Jumbe, Johnstone Kumwenda, Adamson Muula.

**Data curation:** Thokozani Zungu, Shaffi Mdala, Elizabeth Fernando, Halima Twabi.

**Formal analysis:** Thokozani Zungu, Shaffi Mdala, Halima Twabi.

**Methodology:** Thokozani Zungu, Shaffi Mdala, Petros Kayange, Arnold Jumbe, Johnstone Kumwenda, Adamson Muula.

**Supervision:** Petros Kayange, Johnstone Kumwenda, Adamson Muula.

**Writing – original draft:** Thokozani Zungu, Shaffi Mdala, Petros Kayange, Elizabeth Fernando, Halima Twabi, Arnold Jumbe, Johnstone Kumwenda, Adamson Muula.

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
