## [Decision Letter · Decision Letter 0]

10 May 2023

PGPH-D-23-00555

Uptake of diabetic retinopathy screening at a secondary level facility in Malawi

Dear Dr. Zungu,

Thank you for submitting your manuscript to PLOS Global Public Health. After careful consideration, we feel that it has merit but does not fully meet PLOS Global Public Health’s publication criteria as it currently stands. Therefore, we invite you to submit a revised version of the manuscript that addresses the points raised during the review process.

We look forward to receiving your revised manuscript.

Kind regards,

Nasheeta Peer

Academic Editor

Journal Requirements:

2. Please amend your Data Availability Statement and indicate where the data may be found.

Additional Editor Comments (if provided):

Introduction

Lines 64-71: please explain the reason for including a description of UK DR screening programme? Is it relevant to your context?

Lines 71-73: Why describe guidelines for a high-resource setting? Is it applicable to Malawi?

This section is too long; please shorten. The focus should be on Malawi with a clear motivation for conducting this study.

Lines 74-82: This may be incorporated into the Discussion section where it can be compared with the findings in this study. Describing screening in Tanzania and Kenya is not a motivation for this study conducted in Malawi.

Line 81: please improve your grammar – “blinding eye complications”.

Lines 103-105: The aims should be clearly articulated. E.g., “The aim of this study was to…”

Methods

Do patients pay for healthcare?

Does it cost more for DR screening?

Are the waiting times increased if patients undergo DR screening?

Do patients self-select for DR screening?

Are they not automatically provided with an appointment for DR screening?

Statistical analyses: what p-value level was considered statistically significant?

Results

Were only 230 patients seen at the clinic over a 3-month period - October 2020 to December 2020?

Table 1:

What does primary, secondary, etc. schooling refer to? This translates to how many years of schooling completed?

Use the word hypertension, not hypertensive.

Please include HbA1c levels, if available.

Table 1 and 3: please write 1.00 for the reference value for the logistic regression.

Note, this should be presented first.

Where is Table 2?

Discussion

Line 186: name the countries and describe the comparable, or otherwise, prevalence.

Lines 189-192: Please rephrase; there are numerous barriers, not a single one, for the absence of screening programmes.

Lines 203-204: What does “…health promotion on the need for DR screening” actually entail? Please provide more detail.

Why are all patients not routinely referred for DR screening at this diabetes clinic? How are patients selected for DR screening?

Line 220: What do you mean by ‘health care demand’?

Line 222: Uptake of what?

Line 224: replace ‘weak’ with non-significant’. What was the reason for this? Likely the small sample size. Please include.

Lines 232-233: Again, the small sample size with only 49 patients screened is likely the reason for the lack of significance and should be stated.

Line 235: please explain: ’among different races’ – why is race relevant to this study?

Recommendations: Please provide clear recommendations to improve DR at this clinic; these should be linked to your data.

What are areas for further research?

Include the small sample size as a major limitation.

Are there other such studies from Malawi or is it a first? Please include as a study strength.

You did not examine glycaemic control as a factor for DR; this is a limitation. If this data are available, please include in your analyses.

Line 249-254: Please move recommendations and future research to the Discussion section and link it directly to a study finding.

Lines 253-254: “alternative health education models” – the authors need to describe the current education model before advocating for a change.

Reviewers' comments:

Reviewer's Responses to Questions

**Comments to the Author**

1. Does this manuscript meet PLOS Global Public Health’s publication criteria? Is the manuscript technically sound, and do the data support the conclusions? The manuscript must describe methodologically and ethically rigorous research with conclusions that are appropriately drawn based on the data presented.

Reviewer #1: Partly

Reviewer #2: Yes

Reviewer #3: Yes

2. Has the statistical analysis been performed appropriately and rigorously?

Reviewer #1: Yes

Reviewer #2: Yes

Reviewer #3: Yes

3. Have the authors made all data underlying the findings in their manuscript fully available (please refer to the Data Availability Statement at the start of the manuscript PDF file)?

Reviewer #1: Yes

Reviewer #2: Yes

Reviewer #3: Yes

4. Is the manuscript presented in an intelligible fashion and written in standard English?

Reviewer #1: Yes

Reviewer #2: Yes

Reviewer #3: Yes

5. Review Comments to the Author

Reviewer #1: The article addressed a relevant issue that requires more research, especially in low and middle-income countries.

However, the text lacks key contextual information to allow the readers to understand the many barriers people with diabetes face to access timely health care in Malawi.

For example, the article states that the prevalence of DM adults is 2% in rural areas and 3% in urban areas. However, the information comes from a cross-sectional study that, because of the sample size, provides only an estimate, shows that DM is a problem in the country and that there is a high prevalence of risk factors for developing diabetes.

There needs to be more information about Malawi’s health system: how many people have access to health insurance, out-of-pocket expenses, if the country has a DM program and policies, and if the DR screening program offers the exam free of cost. Also, there needs to be more information on the number of clinics that offer the DR screening programs, if these are located in urban or rural areas if there are regions or districts without facilities offering DR screening.

The methods part has to be reviewed and enriched. Authors stated that the protocol adhered to the tenets of the Declaration of Helsinki for research 123 involving human participants, and ethical approval was obtained from the College of 124 Medicine Research and Ethics Committee (study protocol number P.10/19/2803), however, there is no information about the procedure. Informed consent is a process that must respect the autonomy of the participants, and as for this research, recruitment happened within the clinics and has 0 refusal rate; there is a need to know if: recruiters were workers at the clinics, how power relations were addresses, how autonomy was guarantee (beyond the informed consent if for example people were informed about the research in the day of their visit to the clinic, and where invited to come back another day for be part of the research). There needs to be more information about who covered the screening costs. This information is critical to describe how the ethics of the research was guaranteed.

Reviewer #2: The article highlighted the importance of recognizing need for referrals and management for diabetic retinopathy management in underresourced settings. I would also comment if possible on the potential for scalability of this program in-country and methods for scalability and collaboration. It may also be helpful if doing further work after this study to collect information on HgA1c/glucose levels of patients that were referred or need referral for diabetic retinopathy screening. If possible, diabetic teaching and management could also be included during diabetic retinopathy screening as well.

Reviewer #3: Zungu T et al explore a key area in diabetes care of uptake of screening for diabetic retinopathy in a sub-Saharan African country.

Generally, the manuscript is well written with the rationale of the study well explained and the study methods used in data collection plus the exclusion and inclusion criteria well stated.

I have minor comments which the authors should respond to.

1. A more detailed description of the study site is needed. Does it serve a predominantly rural or urban area? How does have an effect on the observed low uptake of the screening services? Kindly elaborate.

2. For the section on predictors of screening services on page 9 and in Table 1 on page 10, add 95% CI to aid in better interpretation of the results.

3. Restrict your p-values to two decimal points apart from the highly significant ones of <0.001.

4. Could the authors also add a note on the availability and cost of treatment options of diabetes retinopathy in that hospital or in Malawi generally (both in the public and private sector) once a patient is diagnosed with severe forms of retinopathy like proliferative diabetic retinopathy.

5. On page 6, the authors highlighted how diabetic screening is done at the different healthcare levels. I would like to know if this screening also includes assessment of other forms of ocular conditions like glaucoma which is also an identifiable cause of blindness in patients with diabetes. If so, how is it done? If it is not done, explain why.

6. PLOS authors have the option to publish the peer review history of their article (what does this mean?). If published, this will include your full peer review and any attached files.

**Do you want your identity to be public for this peer review?** For information about this choice, including consent withdrawal, please see our Privacy Policy.

Reviewer #1: No

Reviewer #2: No

Reviewer #3: No

---

## [Decision Letter · Decision Letter 1]

23 Aug 2023

PGPH-D-23-00555R1

Uptake of diabetic retinopathy screening at a secondary level facility in Malawi

Dear Dr. Zungu,

Thank you for submitting your manuscript to PLOS Global Public Health. After careful consideration, we feel that it has merit but does not fully meet PLOS Global Public Health’s publication criteria as it currently stands. Therefore, we invite you to submit a revised version of the manuscript that addresses the points raised during the review process.

We look forward to receiving your revised manuscript.

Kind regards,

Nasheeta Peer

Academic Editor

Journal Requirements:

Additional Editor Comments (if provided):

Introduction

Lines 99-116: Please shift most of this text on the Malawian context to the Methods section.

Results

The Editor disagrees with the Reviewer on the number of decimal places for p-values when reporting data. Please report all p-values to 3 decimal places in both tables and the main text, as is standard practice.

Discussion

Please improve the syntax of this section. There are many minor grammatical errors that need correction. I suggest a co-author (or anyone else) with an excellent grasp of the English language review this paper in detail for sentence construction, etc.

Lines 290-294: This is an overly long poorly constructed sentence extending over 5 lines with ‘findings’ used twice in the sentence. Please rework.

Lines 294-296: Are the findings from the literature comparable to this study or not? Please comment; that is the purpose of a Discussion.

Lines 297-305: How relevant is the urban-rural comparison to Malawi since most people (84%) live in rural areas? Please edit/rework the text.

Lines 310-314: Please improve the clarity of these sentences as well as the syntax.

Lines 330 and 441: Please replace ‘consumer’ with ‘patient’.

Areas for future research should be ‘suggested’ and not ‘recommended’; please reword accordingly (used in multiple texts). The authors should be more circumspect.

Line 370: replace ‘enough’ with ‘sufficient’.

Lines 370-372: Please improve the syntax.

Line 398: replace ‘condition’ with ‘comorbidity’ in keeping with the context of the paragraph.

Line 402-406: Please explain why a longer duration of diabetes would be associated with DR screening uptake. Simply reporting data from other studies does not contribute to a quality Discussion.

Line 407: Please include the reference for other low-income countries.

Lines 409-411: How does this impact/contribute to DR screening uptake?

Please be consistent with your spelling. I suggest using UK English uniformly e.g., ‘programme’, ‘glycaemic’, etc.

Line 420: ‘Commitment’ in terms of what? (Resources, funding, skills, etc.) Please expand briefly.

Line 424: improve the grammar; this is confusing.

Line 425: Replace ‘studied’ with ‘established’.

Line 429: Reports from which countries? Does it include Malawi?

Line 447: ‘secondary level’ what?

Lines 457-458: Is this feasible/ practical in Malawi where there is likely a shortage of skilled staff who are overburdened with a large volume of patients? The current trend is toward task-shifting i.e., to lower-level health personnel; however, the authors are now suggesting the reverse.

Reviewers' comments:

Reviewer's Responses to Questions

**Comments to the Author**

1. If the authors have adequately addressed your comments raised in a previous round of review and you feel that this manuscript is now acceptable for publication, you may indicate that here to bypass the “Comments to the Author” section, enter your conflict of interest statement in the “Confidential to Editor” section, and submit your "Accept" recommendation.

Reviewer #1: All comments have been addressed

Reviewer #3: All comments have been addressed

2. Does this manuscript meet PLOS Global Public Health’s publication criteria? Is the manuscript technically sound, and do the data support the conclusions? The manuscript must describe methodologically and ethically rigorous research with conclusions that are appropriately drawn based on the data presented.

Reviewer #1: Yes

Reviewer #3: Yes

3. Has the statistical analysis been performed appropriately and rigorously?

Reviewer #1: I don't know

Reviewer #3: Yes

4. Have the authors made all data underlying the findings in their manuscript fully available (please refer to the Data Availability Statement at the start of the manuscript PDF file)?

Reviewer #1: Yes

Reviewer #3: Yes

5. Is the manuscript presented in an intelligible fashion and written in standard English?

Reviewer #1: Yes

Reviewer #3: Yes

6. Review Comments to the Author

Reviewer #1: The authors have addressed my coments regarding context, and ethics.

Reviewer #3: All comments raised have been addressed to my satisfaction.

7. PLOS authors have the option to publish the peer review history of their article (what does this mean?). If published, this will include your full peer review and any attached files.

**Do you want your identity to be public for this peer review?** For information about this choice, including consent withdrawal, please see our Privacy Policy.

Reviewer #1: No

Reviewer #3: No

---

## [Editor Report · Decision Letter 2]

13 Oct 2023

Uptake of diabetic retinopathy screening at a secondary level facility in Malawi

PGPH-D-23-00555R2

Dear Dr Zungu,

We are pleased to inform you that your manuscript 'Uptake of diabetic retinopathy screening at a secondary level facility in Malawi' has been provisionally accepted for publication in PLOS Global Public Health.

Best regards,

Nasheeta Peer

Academic Editor